# Sex Difference and Benzene Exposure: Does It Matter?

**DOI:** 10.3390/ijerph19042339

**Published:** 2022-02-18

**Authors:** Diana Poli, Paola Mozzoni, Silvana Pinelli, Delia Cavallo, Bruno Papaleo, Lidia Caporossi

**Affiliations:** 1INAIL Research, Department of Occupational and Environmental Medicine, Epidemiology and Hygiene Via Fontana Candida 1, 00078 Monte Porzio Catone, Italy; d.cavallo@inail.it (D.C.); b.papaleo@inail.it (B.P.); l.caporossi@inail.it (L.C.); 2Department of Medicine and Surgery, University of Parma, 43126 Parma, Italy; paola.mozzoni@unipr.it (P.M.); silvana.pinelli@unipr.it (S.P.); 3Centre for Research in Toxicology (CERT), University of Parma, Via A. Gramsci 14, 43126 Parma, Italy

**Keywords:** sex, benzene, biomarkers, in vivo studies, epidemiological surveys

## Abstract

Sex-related biological differences might lead to different effects in women and men when they are exposed to risk factors. A scoping review was carried out to understand if sex could be a discriminant in health outcomes due to benzene. Studies on both animals and humans were collected. In vivo surveys, focusing on genotoxicity, hematotoxicity and effects on metabolism suggested a higher involvement of male animals (mice or rats) in adverse health effects. Conversely, the studies on humans, focused on the alteration of blood parameters, myeloid leukemia incidence and biomarker rates, highlighted that, overall, women had significantly higher risk for blood system effects and a metabolization of benzene 23–26% higher than men, considering a similar exposure situation. This opposite trend highlights that the extrapolation of in vivo findings to human risk assessment should be taken with caution. However, it is clear that sex is a physiological parameter to consider in benzene exposure and its health effects. The topic of sex difference linked to benzene in human exposure needs further research, with more numerous samples, to obtain a higher strength of data and more indicative findings. Sex factor, and gender, could have significant impacts on occupational exposures and their health effects, even if there are still uncertainties and gaps that need to be filled.

## 1. Introduction

The consideration of sex and gender as parameters linked with different health outcomes is a mandatory challenge in all the different specializations of modern medicine [1]. The scientific path that led to this consideration is to be considered a turning point in the approach to medicine and advancement in our knowledge [2].

Sex-related biological differences might lead to different effects in women and men when exposed to risk factors [3,4,5,6].

The Committee on Understanding the Biology of Sex and Gender Differences of the US Institute of Medicine distinguished sex and gender with the following definitions: “sex is the classification of living things generally as either male or female, according to their reproductive organs and functions assigned by the chromosomal complement” while gender is “a person’s self representation as male or female, or how that person is responded to by social institutions …” [7].

Too many studies, even in occupational settings, were conducted on male sample excluding female subjects [8]. In particular, the issue of sex and gender was introduced in work-related wellbeing in 1995 in the World Conference of Beijing, and in 1999 the International Labour Office [9] took it up. Recent research [6,10,11] showed that male workers were exposed to more chemicals and physical exertion than females; however, for many of these occupations, the presence of women was not irrelevant. Women had a predominance of activity with repetitiveness and rapidity [10], in addition to specific chemical exposures. However, considering the same occupational settings and the same tasks, evidence emerged of a higher risk of all types of injuries (OR = 1.365, 95% CI 1.290–1.445) and acute injuries (OR = 1.201, 95% CI 1.151–1.295) for female workers, for example among aluminum smelter workers [12], than male ones.

With regard to chemical susceptibility, biological differences could lead to different health outcomes [13,14]. Between sexes, some features need to be considered such as body size and composition, hormonal asset, the rate of specific mass (muscle vs. fatty vs. bone) [15]. In particular, a greater amount of adipose tissue might make women more susceptible to lipophilic substances (such as benzene) [15]. Most of what we know about chemical toxicity for humans comes from occupational exposure studies and epidemiological investigations. The Scientific Group on Methodologies for the Safety Evaluation of Chemicals organized a workshop on “gender and toxicology” [16] the result of which was entered in the *Framework for Gender Differences in Human and Animal Toxicology* [17], where it was underlined that sex was not adequately considered in toxicology and epidemiology.

In fact, both toxicodynamic and toxicokinetic actions are subject to modification with regard to sex elements (gonadal steroids, growth hormone, agonists or antagonists of receptors) [18]. Generally, the adverse health effects vary depending also how quickly and efficiently chemical agents are metabolized [18]. Furthermore, it is important to note that sex could influence toxicokinetic phases [19].

Clewell et al. [20] reviewed the literature on toxicokinetics regarding male/female similarities and differences. He highlighted that in the absorption phase, there were no significant differences in case of oral absorption (both for lipophilic and hydrophilic agents); the same findings emerged in the case of inhalation for lipophilic agents and particulates, while for the inhalation of hydrophilic chemicals and dermal absorption the data were insufficient. The distribution phase showed a higher rate in females for lipophilic chemicals, and a higher rate for males in the case of hydrophilic molecules and protein-bound molecules.

Xenobiotic blood concentrations depend on the capacity of distribution and amounts of renal clearance: usually, men have higher blood volume while women have higher fat content [21]. Females have a minor glomerular filtration rate (lower average renal clearance), which results in a slower elimination from the body, or a higher deposition in repository organs [21]. Evidence about metabolizing phase underline a higher use of glucuronyl transferase and P450 system by males than females, while data about other means of metabolism were insufficient (glutathione S-transferase, sulfotransferase, carboxylesterase, alcohol dehydrogenase) [20]. Finally, data about the elimination phase were less numerous, but considerations about differences in glomerular filtration and renal clearance led to caution [20]. Nevertheless, there are still uncertainties about the role of sex differences in toxicology [19].

Benzene is a highly volatile compound; human exposure is due to environmental contamination particularly in occupational settings, and it occurs mainly through inhalation [22]. Activities with benzene use are: the oil industry, coal coking plant, chemical industry. It is a component of petrol (gasoline) and heating oils and used as a chemical intermediate in some consumer production [23]. Other causes of benzene in the atmosphere are automobile exhaust gases, off-gassing from building materials, industrial discharge, landfill leachate and cigarette smoke [23]. The International Agency for Research on Cancer (IARC) has recorded benzene in group 1, considering the literature evidence of its carcinogenic effects in humans [24]. Benzene is a possible cause of acute myeloid leukemia, chronic lymphocytic leukemia, aplastic anemia, myelodysplastic syndrome, but also other disorders [25].

In fact, epidemiological studies highlighted developmental effects (e.g., increase of offspring with spina bifida and alteration in several fetus parameters), reproductive effects (e.g., alteration of sperm characteristics), immunological effects (e.g., allergy, sensitization, increases in immune cells), effects on respiratory function (e.g., asthma, cough, wheeze, odds of pulmonary infection and pulmonary inflammation, odds of bronchitis risk, decrease in pH of exhaled breath condensate), cardiovascular disease and metabolic effects (e.g., metabolic syndrome, insulin resistance, diabetes mellitus) [26,27,28,29,30,31].

In in vivo studies (rats and mice) [25,32,33,34], this substance showed carcinogenic activity, and genotoxic and hematotoxic effects were documented. Several reasons for benzene-induced neoplasia, linked to genotoxic activity, have been suggested: (1) reaction of benzene metabolites with DNA producing DNA adducts; (2) oxidative DNA damage; (3) aneugenic or/and clastogenic effects [24,35].

The metabolism of benzene is complex and consists of many possible pathways (Figure 1). Benzene is metabolized principally by cytochrome P450 2E1 (CYP 2E1) in a reactive intermediate, benzene oxide [36]. Microsomal epoxide hydrolase (mEH) converts this intermediate to benzene dihydrodiol, which is then converted to catechol by a dehydrogenase [37,38]. Benzene oxide can also rearrange forming phenol, which then reacts with little amount of cathecol, and by CYP 2E1, produces hydroquinone. These chemicals are substrates for further reactions of oxidation that produce 1,4-benzoquinone and 1,2-benzoquinone. These two molecules are considered as toxic metabolites of benzene, owing to their ability to react with macromolecules and lead to toxic effects. NAD(P)H quinone oxidoreductase-1 (NQO1) catalyzes the inverse reaction with a detoxifying effect. A smaller amount of benzene oxide can be metabolized forming muconic acid or phenyl mercapturic acids [38]. The balance of benzene metabolism and the relative amount of the different metabolites are the key information to understand possible genotoxic effects. Therefore, understanding the role of involved enzymes is critical to obtaining a lower or higher toxicity [37,39].

To predict the risk of benzene exposure related to sex differences a study employed physiologically based pharmacokinetics modeling [41]. Women showed a higher speed of metabolization than men, however female subjects generally had a higher body fat rate and this factor influenced the internal dose. The findings demonstrated that, even considering similar exposure situation, benzene blood concentrations were usually greater in men, but women metabolized more benzene than men (+23–26%) at the same time [41]. These results suggest that women and men could have different health risks owing to benzene exposure and that risk assessment of exposure standards based on the male sample might not be protective for the female population.

Considering the experimental results, the United States Environmental Protection Agency (US-EPA), suggested different health effects of benzene based on sex differences, even if with mixed results [30,42].

The aim of the study herein presented is to perform a scoping review in order to clarify the possible role of sex in the health effects due to benzene exposure, starting from animal models to epidemiological surveys.

## 2. Materials and Methods

### 2.1. Literature Search Methodology

This review of scientific literature was realized following the “PRISMA” statement [43], (guidelines for reporting systematic reviews and meta-analyses).

The literature papers were gathered up to 31 December 2022, via the two electronic bibliographic databases Pubmed and Scopus.

The keywords used for the searching scheme (in title or abstracts) were an association of “benzene” and “gender”, “sex”, “gender difference”, “sex difference”, “man”, “women”, “male” and “female”.

### 2.2. Eligibility Criteria and Study Selection

Only studies published in Italian, Spanish or English were included. Papers recording original information regarding the exposure to benzene and sex/gender differences, both in animals and humans, were considered.

The exclusion criteria provided for the elimination of in vitro studies, reviews and meta-analyses, together with articles that did not consider both sexes.

The selection of only scientific papers that considered both sexes, instead of studies evaluating only one sex, reduces the number of the selected articles, but on the other hand reduces the bias of the results, because males/females are in the same experimental conditions.

Figure 2 shows schematically the selection process of the papers.

Two researchers conducted the eligibility assessment of the studies autonomously. Divergences were resolute by consensus. Two authors, independently, evaluated results of each paper, including study quality indicators.

In vivo studies considered males and females both exposed, different route of exposure (inhalation, oral, gavage, intravenous or intraperitoneal), and time of exposure (from acute to chronic). The comparison of the articles selected was based on findings and different analytical methods were not considered. Considering that almost all of these works were carried out on mice and a few rats, studies on other animals were excluded.

The epidemiological studies considered were: (1) about air pollution exposure or cigarette-smoking habit; (2) retrospective studies, usually case/control, but also cross-sectional ones, with a focus on specific illness and surveys on possible occupational risk factors; (3) studies of exposure to benzene in occupational settings.

The assessment of bias within studies was carried out using the Cochrane risk of bias tool [44]. Considering the different types of bias presented by the tool, we took into consideration the last three: incomplete outcome data, selective reporting and “other bias”. The Cochrane tool identified also other types of bias, but these are related to clinical trials, so they are not applicable for the studies collected in this paper.

In “other bias”, the sample size, the number of controls presented, and the data source were considered in human studies, while the sample size, the route and time of exposure, and the species and strain were considered in in vivo studies.

## 3. Results

The findings that emerged from experimental studies and from human epidemiological surveys were collected and summarized in the following paragraphs. Overall, the endpoints were very different, sometimes for the same endpoint different analytical approaches were used. Nevertheless, the findings must be read in terms of their single outcomes and specific significance, and not necessarily in terms of a comparison among studies.

### 3.1. Sex Difference and Benzene Exposure: Data on Animals

Table 1 shows the most significant results related to in-vivo benzene exposure performed in animals of both sexes, particularly mice and rats.

#### 3.1.1. Genotoxicity and Hematotoxicity

Genoxicity and hematoxicity due to in vivo benzene exposure by inhalation is well known and some studies reported consideration about sex differences. Benzene is clastogenic and aneugenic inducing sister chromatid exchanges, chromosomal aberrations, and micronuclei formation. Bone marrow, along with spleen and liver, is its main target [35]. The measurement of the frequencies of micronucleated polychromatic erythrocytes (MN-PCE) allows an evaluation of bone-marrow damage [47], while the percentage PCE in peripheral blood permits the alterations in the rate of erythropoiesis to be assessed [47,49].

Tice et al. highlighted how acute exposure of adult mice significantly delayed production of bone marrow cells only in males [32]. Luke et al. [47] described a marked higher frequency of MN-PCE, five-fold greater in male mice compared to females. Similarly, Tice et al. [49] studied the magnitude of genotoxic damage due to benzene, considering in particular the route, regimen and duration of exposure, in mice of both sexes, of different strains. This work highlighted that the increase in MN-NCE and MN-PCE depended on the sex (male > female) and on the strain of mice, and only for MN-PCE on the route of exposure (inhalation > oral). Additionally, exposure induced a significant reduction sex-dependence (male > female) in the percentage of PCE in the peripheral blood, packed cell volume, and the cellularity in the tibia bone marrow.

Haemotopoiesis occurs mainly in the bone marrow [55], and hence is affected by benzene exposure [35]. Observing effects on mice after inhalation of benzene, Corti et al. [50] found a higher susceptibility of male erythron; in fact a reduction of erythroid colony-forming units (CFU-e numbers) was highlighted in male mice, and no changes were observed in females. Faiola et al. [25], investigated the sex-specific gene expression induced by benzene exposure, and it emerged that the mRNA levels of 18 genes, determined in HSCs from male and female mice, presented sex differences for some genes.

Similar results were reported after benzene oral exposure. In research of Gad-El Karim et al. [45,46] it emerged that after two acute oral exposures, male mice had higher induction (from two to three times) of both micronuclei (MN) and chromosomal aberrations than females. Huff et al. [48] reported quite a prevalence in male mice and rats of carcinogenic effects.

However, not all studies agree with regard to sex-based differences about the genotoxicity and hematotoxicity of benzene. Giver et al. [34] studying the possible persistence of aneuploid cells in hemopoietic sub-populations identified no differences between male and female groups even after 8 months.

Effects following prenatal benzene exposure of mice have also been described. Transplacental exposure affected fetal metabolism differently according to sex: hydroquinone and cathecol concentrations were higher in male and female fetuses, respectively. The same study showed an increasing trend of tumors in offspring: hepatic tumors were higher in males, while hematopoietic tumors in females [52]. In addition, increased oxidative stress and micronuclei formation, and alterations in signaling pathways involved in normal hematopoiesis, were observed, with a higher susceptibility in male mice with respect to females [56].

#### 3.1.2. Metabolism and Metabolic Effects

To investigate the sex difference in metabolism, several in vivo studies were performed exposing animals not only to benzene but also to their metabolites.

Kenyon [57] observed differences linked with sex, in the urinary excretion of phenol metabolites, in particular the major oxidized metabolite of benzene, after intravenous or gavage ^14^C-phenol exposure of mice. Male mice produced more hydroquinone (HQ) glucuronide than females, this brought researchers to hypothesize that the oxidation of phenol to hydroquinone (HQ) is faster in male mice. The same authors [33], observed a sex difference in rates of metabolism measuring the elimination of benzene, phenol and HQ from the blood, following benzene exposure by inhalation; the elimination of phenol from the blood was more rapid in male mice, suggesting a higher speed of metabolization for males. These data were confirmed in a later paper which found a correlation between the maximum rate of oxidation in male mice and concentrations of hydroxylated benzene metabolites in tissues and water-soluble metabolites in urine (HQ glucuronide and muconic acid) [58].

Several authors investigated the role of detoxifyng enzymes in benzene metabolism. Bauer et al. [37] studied the importance of NAD(P)H quinone oxidoreductase-1 (NQO1), the enzyme detoxifying benzoquinones using mice, of both sexes, NQO1 deficient (NQO1−/−) and mice with wild-type NQO1 (NQO1+/+). Micronucleated peripheral blood cells (MN-RET and MN-NCE) were used as an effect indicator of genotoxicity while for evaluating hematotoxicity and myelotoxicity the peripheral blood counts and bone marrow histology were used. Furthermore, for the assessment of possible DNA damage p21 mRNA levels in bone marrow cells were evaluated. Results indicated that NQO1 is a central enzyme for detoxification action, in female animals, both for genotoxicity and hematotoxicity; while in male animals it is important only for the detoxifying activity for hematotoxicity. The same year, they studied [39] the role and the sex difference related to another enzyme involved in benzene toxicity: the microsomal epoxide hydrolase (mEH). The results highlighted that mEH is important in the process of toxicity of benzene (both on genes and on blood cells) and induction of the p53 DNA damage response in male, but not in female, mice.

Debarba et al. [51] studied how benzene could affect the metabolic balance. Focusing on glucose homeostasis, peripheral lipid metabolism and hypothalamic inflammation and neuroinflammatory, they found that chronic exposure to benzene by inhalation at low levels induced health effects linked with a metabolic imbalance and also linked with sex. Notably, exposure promoted hyperglycemia and hyperinsulinemia only in male mice. Additionally, in livers, the expression of genes associated with possible metabolic alterations resulted significantly increased only in male mice, and similar the serum concentrations of triglyceride, low-density lipoprotein (LDL) and non-esterified fatty acid (FFA) were also higher in male mice. Finally, researchers suggested that benzene produced hypothalamic inflammation and neuro-inflammatory responses in male mice. In fact, the number of microglial and astrocytes in the hypothalamus increased significantly in exposed male mice, while no differences were observed in female ones.

With regard to prenatal exposure, Koshko et al., demonstrated that benzene induce metabolic perturbation (energy homeostasis and glucose metabolism), more severe in male offspring than in females [53].

Considering the findings of the various studies, presented in Table 1, it is possible to summarize the elements related to the sex difference as shown in Table 2.

### 3.2. Sex Difference and Benzene Exposure: Findings in Humans

A recent investigation of scientific literature [59] underlined that, with regard to the published papers on benzene exposure, only 14.2% of these studies were obtained from both male and female subjects and among these only 16.3% were epidemiological surveys. This demonstrates how few human investigations are currently available.

Table 3 shows the main outcomes of the studies in humans related to benzene exposure and consideration sex differences.

The studies oriented to highlight the hematotoxicity of benzene showed a higher vulnerability of women instead of men. In female workers exposed to benzene [62], abnormal levels of blood cells (white blood cell, neutrophil count and platelets) were found compared with male workers, and the used biological indexes of benzene exposure (S-phenylmercapturic acid, 8-hydroxy-2′-deoxyguanosine and malondialdehyde) were significant higher (respectively +555%, +183% and +33.3%) in women.

Poynter at al. [60] investigated the role of exposure to different chemicals in the onset of acute myeloid leukemia (AML) and myelodys-plastic syndromes (MDS). It emerged, for AML, that the benzene exposure was clearly linked with an increased risk, and in particular, the stratification by sex for females had almost a double OR than for males (2.84, 95% CI 0.96–8.39 vs. 1.60, 95% CI 1.04–2.45) even with a low statistical significance.

However, other researchers disagree with these results and a study of 74.828 benzene exposed workers did not find a sex-related higher incidence of leukemia and hematopoietic and lymphoproliferative disorders [65]. Low-dose exposure brought no significant difference in blood benzene concentrations between sexes [64], while changes were observed in the urinary leukocyte count, more significant in females than males. The higher values of benzene in women were linked to increased urinary white blood cells by 2.902%.

Regarding the genotoxicity less evidence emerged about possible sex differences and no statistical differences between sexes were found in genotoxic biomarkers (micronuclei assay, comet test) [63], while women exposed to BTEX showed a higher risk of chromosomal abnormalities, even if with a weak statistical significance [66].

The only mechanistic survey [41] demonstrated that, even considering similar exposure situation, women metabolized more benzene than men (+23–26%) at the same time, so blood benzene was higher in men. These findings seem to confirm other data [63] showing that even with similar airborne benzene levels, female workers had increased median t,t,muconic acid/creatinine concentrations compared to the male group.

Furthermore, evaluating the cardinal symptoms of exposure to benzene, and other volatile compounds, women showed five or more symptoms, with greater probability (more than 50%) than men [61].

The main elements related to the sex difference are summarized in Table 4.

## 4. Discussion

Many studies on the effects of in vivo benzene exposure have been reviewed, but few take the sex differences into consideration. In addition, most of this research is from the 1980s and 1990s, although in recent years the interest in this field has been increasing.

The toxic effects of benzene in animals are widely known [25,49,50,67,68]. The degree of the effects is linked to the level of exposure, strain, and animals involved [24,48,49,50,51,54,57], but also to sex. With regard to sex differences, male mice and rats of multiple strains seem to be more vulnerable to genotoxic and hematotoxic effects of benzene than females. Although with different intensity, this aspect is also evident with different exposure protocols (e.g., time and route of administration) [33,45,46,47,57].

Genotoxic and hematotoxic effects were observed by evaluating induction of MN in both bone marrow PCE and peripheral blood erythrocytes, as well as in the decrease in WBC and in HSCs or the depression in the rate of erythropoiesis as measured by the percentage of PCE in peripheral blood. Cytotoxic damage (% PCE, PCV, cellularity) higher than in exposed females was also evident [49]. The peripheral blood MN analysis provides useful information because it allows a concurrent evaluation within the same animal of both acutely induced and chronically accumulated bone marrow damage. Thus, this assay is ideally suited for evaluating the exposure duration-dependent alterations in bone marrow vulnerability to genotoxic chemicals. By evaluating the frequency of MN-PCE, the link between benzene exposure duration and genotoxic damage could be not so clear, owing to possible confounding factors such as the rate of erythropoiesis [49].

Hematopoietic stem cells (HSCs) could experience changes in their structure owing to DNA damage following benzene metabolites action [25]. Notably, as regards the study of Faiola et al. [25], the higher level of *bax* mRNA together with the absence of differences in bcl-2 levels in male mice suggest an enlarged vulnerability to apoptosis in male rather than female mice. In females, a significant decrease in mRNA levels for *p53* and *mdm-2*, a gene under the control of *p53*, indicates the suppression of p53 function. These results may partially explain the very low level of myelotoxicity and hematotoxicity observed in female mice compared with males after benzene exposure [25,69].

Faster benzene oxidative metabolism in males compared to females correlates with their greater sensitivity to develop disorders after in vivo exposure [34]. In fact, metabolism is deeply involved in benzene toxicity since inhibition and induction of oxidative metabolism, respectively, mitigate and enhance these effects [58].

Myelotoxicity and hematotoxicity due to benzene are linked to the Cyp2e1 activity, and the production of reactive intermediates of benzene [51]. Furthermore, males excreted a higher proportion of hydroquinone glucuronide than females and the hypothesis is that male animals produce hydroquinone faster than females [57]. Finally, males have a higher maximum rate of benzene oxidation and a faster disappearance of phenol from the blood [33]. These results are consistent with both the higher genotoxicity of hydroquinone versus phenol and the higher vulnerability of males, registered with MN and SCE.

Similarly, the role of NQO1 and mEH enzymes is critical; the first sustains the reduced form of quinones and their derivatives so that they can react to easily form conjugated molecules and can be readily expelled while the latter is involved in the detoxification of toxic epoxides [37,39]. Bauer et al. [37,39] found a sex difference in the metabolic role of NQO1 and mEH, highlighting their importance with respect to benzene-induced toxicity. In fact, NQO1 has a detoxifying role, more evident in female mice (both for genotoxicity and hematotoxicity) while in male mice the effect is clear only for hematotoxicity. Instead, mEH is a key enzyme for male mouse benzene-induced toxicity while no evidence emerged for females. These findings suggest sex-related differences in benzene metabolism and toxicity.

Debarba et al. [51] demonstrated that benzene causes a different metabolic imbalance (glucose homeostasis, peripheral lipid metabolism and inflammation of the hypothalamus and neurological) in terms of sex-related differences. The reason why is not clear; and some authors [25] suggested a possible involvement of male hormones and sex-related differences in benzene metabolism could translate into different effects.

Prenatal exposure in in vivo experiments demonstrated that benzene could cross the placenta and reach the fetus, and different sex-based effects were observed [53]. In agreement with the greater susceptibility of males to benzene effects, male fetuses appeared more vulnerable to oxidative stress caused by benzene exposure [56] while adult male offspring developed insulin resistance [53]. In addition, offspring showed a trend of higher development of hepatic tumors in males and hematopoietic tumors in females. The different metabolite concentrations (catechol and hydroquinone) found in fetuses of males and females might affect the development of hepatic and hematopoietic fetal cells [52]. This result is line with human data: liver cancer is more prevalent in male children [70], while female infants show higher incidence of leukemia [71]. Beyond these experimental results, the precise mechanism involved in benzene toxicity and induced-carcinogenesis due to prenatal exposure are still unknown [56].

Regarding evidence in humans, sex-specific divergences seem to be a parameter of interest considering benzene exposure and its health effects. The published findings showed a possibility of greater risks of specific health effects for women, such as leukemia, although with some uncertainties [64], epidemiological evidence shows the greater involvement of women in adverse effects related to exposure to benzene, especially considering the hematochemical profile [62], while regarding other possible effects, like genotoxic effects, weak association emerged [66].

There was an increased interest over the years for non-cancer effects of benzene, in particular dismetabolic diseases like metabolic syndrome [72], insulin resistance in children/adolescents [28,29], and elderly adults [30] with type 2 diabetes [28,73]. However, in most cases, these articles are not focused on the difference between males and females or describe non-conclusive/preliminary results. In addition, it is not clear how benzene contributes to all these health effects [26,31]: few hypotheses have been proposed and they rarely consider the difference between sexes.

One hypothesis [26] considered that benzene interferes with the pathways and/or the mechanism of action of endogenous hormones. However, even if benzene is suspected of having endocrine-disrupting properties, there exists a controversy [74], and without doubt it is documented that it affects reproductive organs but no estrogenic, androgenic or steroidogenic activities were highlighted.

Health effects owing to benzene exposure were proposed also in the case of low levels of concentration, as in environmental air [27,28], with greater susceptibility during human development [75]. Therefore, it is important to evaluate the effect of benzene exposure considering the various stages of human life: pregnant women, children, adults, and the elderly as well as prenatal exposure (from in vivo experiments). Even preconception and prenatal exposure to high ambient chemical agent levels, including benzene, have been related to gestational diabetes mellitus [31].

However, these studies did not report differences between sexes, with the exception of Choi et al., who observed a significant dose-dependent relation between insulin resistance and benzene in women, even if men showed stronger effects [30].

Furthermore, it must be taken into account that benzene as an environmental contaminant is often correlated to other xenobiotic pollutants (e.g., nitrogen oxides, other aromatic compounds, particulate matter) associated with the development of chronic diseases, which makes the interpretation of the benzene exposure effect and the risk of associated health effects more difficult [31]. In addition, benzene, as with other aromatic hydrocarbons, can originate in other chemicals that can also act as endocrine disruptors [26].

With regard to benzene exposure and metabolic effects, the precise mechanisms remain unknown and we can only speculate. One hypothesis is that the excessive oxidative stress and/or systemic inflammation caused by benzene might lead to pancreatic β-cell dysfunction and insulin resistance which, in turn, may play a crucial role in the development of metabolic syndrome [72] and/or other chronic conditions including diabetes mellitus (DM) [28,30,31,73] and cardiovascular disease [28]. In fact, urinary t,t-muconic acid concentration has been positively associated with an increased cardiovascular disease risk, even if there is no difference between sexes [28] and the levels of oxidative stress biomarkers, such as malondialdehyde and/or superoxide dismutase [28,30]. However, t,t-muconic acid metabolite can also arise from sorbic acid, a food additive, and hence its exclusive use to assess benzene exposure can lead to misinterpretation [30,76]. Finally, evidence that emerged about a different rate of metabolization of benzene in women rather than in men [41] calls for further biochemical surveys to understand how this difference could lead to toxicologically relevant effects.

Even if our literature search was very detailed, there are a few limits. First, the sample size is critical to assess the strength of the results that emerged. Studies involving a few dozen subjects present evidence with an inevitably very low statistical strength and therefore the extension of the data to a general population is difficult. A separate case are the mechanistic biochemical investigations, in which the data even on a few subjects can lead to significant results and information. The possible risk of inaccuracy in reported studies [77] could be an overestimation of sex difference compared to the consideration of other factors. In particular, anatomical or physiological factors could represent significant variables; for example, the mass difference (small men vs. large women) in terms also of the fat composition, and indeed fat percentage is different between males and females. Other factors are age and physical activity, and researchers rarely collect all these data. Finally, in in vivo studies, the different analytical methods were not considered because the comparison of the articles selected was based on findings, the papers being very different, from the study design to the outcomes. However, the interest of this review is not in the single analytical data but consideration of the differences between sexes. For this reason, the only way to avoid bias arising from different methodological approaches is considering the findings of each study.

Regarding animal data, it seems that male mice and rats of multiple strains are consistently more sensitive to benzene-toxic effects than females. The faster benzene oxidative metabolism in males is in agreement with their greater sensitivity to developing disorders after in vivo exposure. On the other hand, epidemiological studies showed different results. In particular, women seem to be at higher health risk than men, even at the same exposure levels, probably linked to a different time of metabolization, a higher rate of accumulation in fat tissues, and a more consistent blood susceptibility. This opposite trend highlights that the extrapolation of in vivo findings to human risk assessment should be taken with caution. The analysis of findings on rodents could be of certain interest, but data must be given the right attention, because the specific characteristics of the single molecule are linked with extrapolative factors or failure [78].

## 5. Conclusions

When talking about occupational exposure to chemicals, sex is without doubt a parameter of certain influence, in term of toxicokinetic and toxicodynamic effects of the toxins in the male or female body. Nevertheless, even when doing the same type of work, men and women could be engaged in different activities and therefore they could be exposed to various situations in workplaces, so the sex element can play a role [11]. Sex and gender could have important influences on work-related exposures and health outcomes and, undoubtedly, there are gaps in our knowledge concerning these influences. Systematic reviews showed, for benzene, that only 14.2% were sourced from studies using both sexes, only 16.3% had epidemiological data, 19.2% of in vivo data and 4.8% of in vitro results [59].

The present scoping review regarding benzene exposure and possible differences in health effects between the sexes, seems to suggest a significant role for this parameter, looking at the toxicokinetic and toxicodynamic considerations. Nevertheless, uncertainties exist, particularly observing the controversies among the findings on animals and on humans.

The topic of sex and gender differences, linked to benzene exposure, needs further research in humans, with more numerous samples to obtain a higher strength of data and more indicative findings. The evidence calls for particular attention of the occupational physician, during health surveillance in workplaces, to give sex the right consideration with regard to specific health effects and different possible incidences of disease.

## Figures and Tables

**Figure 1 ijerph-19-02339-f001:**
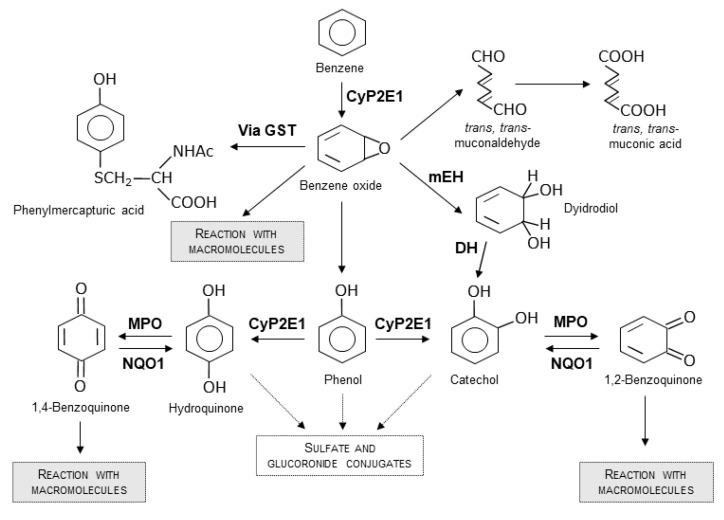
Simplified metabolic scheme for benzene showing major pathways. Adapted from National Toxicology Program, 1986 [40] and from Bauer et al., 2003 [37,39]. Abbreviations. CyPE1: cytochrome P450 family 2 subfamily E; GST: Glutathion-S-transferase; mEH: microsomal epoxide hydrolase; DH: benzene dihydrodiol dehydrogenase; MPO: myeloperoxidase; NQO1: NAD(P)H quinone oxidoreductase-1.

**Figure 2 ijerph-19-02339-f002:**
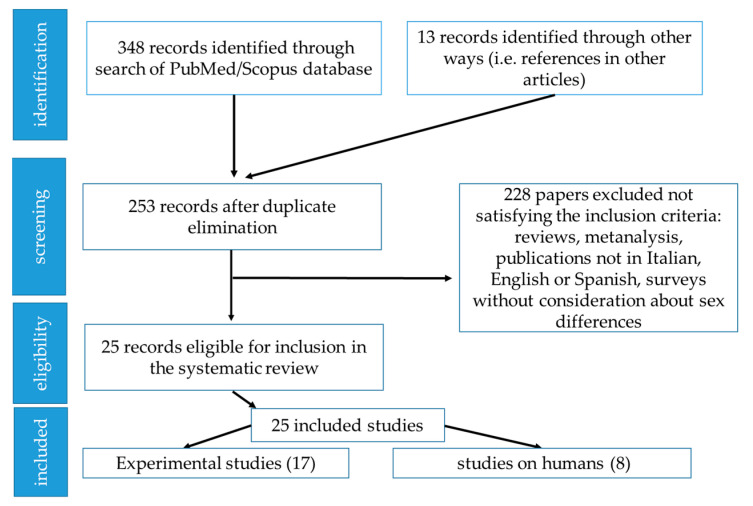
Selection process for articles.

**Table 1 ijerph-19-02339-t001:** The main outcomes of in vivo studies following benzene exposure in both males and females.

Reference	Substance and Type of Exposure	Exposure Conditions	Species and Strain	Main Endpoints	Outcome Males vs. Females	Interest of the Study
Tice et al., 1980 [32]	Benzene Acute exposure	Exposure by inhalation: 3100 ppm for 4 h	DBA/2 mice	SCE ^1^ BM ^3^ cellular proliferation frequency of CA ^4^	SCE: n.s. ^2^ BM cellular proliferation: significantly decreased (*p* < 0.01) frequency of CA: n.s.	Benzene exposure significantly delayed proliferation of BM cells only in males
Gad-El Karim et al., 1984 [45]	Benzene Acute exposure	Oral exposure: 440 mg/kg for 24 h	CD-1 mice	MN ^5^ in BM ^3^ frequency of CA in BM	MN in BM: increased frequency of CA ^4^ in BM: increased	Female mice were more resistant to the genotoxic effect following benzene exposure
Gad-El Karim et al., 1986 [46]	Benzene Acute exposure	Oral exposure: 220, 440, and 880 mg/kg	CD-1 mice	MN in BM unconjugated PHE ^6^	MN in BM: increased unconjugated PHE ^6^: decreased	Benzene myeloclastogenicity was linked to its metabolism
Luke et al., 1988 [47]	Benzene Sub-chronic exposure	Exposure by inhalation: 300 ppm, 6 h/day, 1–5 days/week for 13 weeks	DBA/2 mice	MN-PCE ^7^ MN-NCE ^8^	MN-PCE: significantly increased (*p* < 0.01) MN-NCE: n.s.	Benzene showed higher genotoxicity in male mice than female, being a five-fold greater frequency of MN-PCE in males
Huff et al., 1989 [48]	Benzene Chronic exposure	Oral exposure: Male rats 0, 50, 100, or 200 mg/kg bw, 5 days/week, for 103 weeks Female rats 0, 25, 50, or 100 mg/kg bw, 5 days/week, for 103 weeks. Male and female mice 0, 25, 50, or 100 mg/kg bw, 5 days/week, for 103 weeks	F344/N rats B6C3FW mice	Incidences of neoplasms	F344/N rats Higher presence of squamous cell papillomas and squamous cell carcinomas B6C3F1 mice Higher presence of Harderian gland adenomas	Clear evidence of carcinogenicity of benzene in both sexes of rats and mice with some prevalence in male.
Tice et al., 1989 [49]	Benzene Sub-chronic exposure	Exposure by inhalation: 300 ppm, 6 h/day, for 13–14 weeks Oral exposure: 80 mL/kg by gavage, 5 days/week, for 14 weeks	DBA/2 mice B6C3Fj mice C57BL/6 mice	MN-PCE MN-NCE % PCE ^9^ PCV ^10^ BM cellularity	MN-PCE: significantly increased (*p* < 0.0001) MN-NCE: significantly increased (*p* < 0.0001) % PCE: significantly decreased (*p* < 0.05) PCV: significantly decreased (*p* < 0.025). BM cellularity: significantly decreased (*p* = 0.0002)	Evaluation of genotoxic and cytotoxic effects owing to benzene, considering specific factors: sex, dose, strain and animal
Kenyon et al., 1996 [33]	Benzene Acute Exposure	Exposure by inhalation: Low: 400–500 ppm, Intermediate: 1200–1300 ppm, High: 2600–2800 ppm	B6C3F1 mice	Vmax ^11^ Blood PHE ^6^ PHE ^6^ t_1/2_ ^12^	Vmax ^11^: Increased Male: 14.0 mmol/h-kg Female: 7.9 mmol/h-kg Blood PHE ^6^: significantly decreased (*p* = 0.012) PHE ^6^ t_1/2_ ^12^: significantly decreased (*p* < 0.012) Male: 2.44 (1.97, 3.19) Female: 3.43 (3.08, 3.89)	Male mice showed a higher maximum rate of benzene oxidation and a faster disappearance of phenol from the blood compared with females
Corti et al., 1996 [50]	Benzene Sub-acute exposure	Exposure by inhalation: 10 ppm for 6 h/day, for 10 days	Swiss Webster mice	N° CFU-e ^13^ in BM ^3^ cells	N° CFU-e ^13^ in BM ^3^ cells: decreased in males	Investigating how/if sex may affect depression in CFU-e numbers
Giver et al., 2001 [34]	Benzene Acute exposure	Oral exposure: 220–880 mg/kg, for 2–4 days	C57Bl/6J mice	Persistence of aneuploidy in primitive hemopoietic cells	n.s. ^2^ (Up to 14% aneuploid cells in males and females)	The effect of Benzene on DNA of immature/primitive cells persists both in male and female.
Faiola et al., 2004 [25]	Benzene Sub-acute exposure	Exposure by inhalation: 100 ppm for 6 h/day, 5 days/week, for 2 weeks	129/SvJ mice	genes from HSCs ^14^ isolated from BM: *p21*, *bax*, *wig1*, *p53*, *gadd45a*, *ku80*, *ccng*, *prkdc*, *rad51*, *rad54*, *rpa*, *apex1*, *pcna*, *DNAPolβ*, *xpc*, *xpg*, *bcl-2*, *cyclin G*, *mdm-2*, and *gapdh*	Male vs. Female *ku80*, *ccng* and *wig1*: significantly higher (*p* < 0.05)	The gene expression profiles may partially explain the sex-related differences in hematotoxicity and myelotoxicity.
Debarba et al., 2020 [51]	Benzene Sub-acute exposure	Exposure by inhalation: 50 ppm for 6 h/day for 4 weeks	C57BL/6 mice	Blood glucose Blood Insulin *Cyp2e1* ^15^ Genes associated with gluconeogenesis: *G6pc* and *Ppck1* Genes associated with lipid and fatty acids synthesis: *Srebp1*, *Srepb2*, *Abca1*, *Cpt1*, and *Acc* blood serum levels of triglyceride, LDL ^16^ and FFA ^17^ hepatic inflammatory genes: *Il1* and *Il6* N° of microglia and astrocytes	Blood glucose: higher (*p* < 0.05) Blood Insulin: higher (*p* < 0.05) *Cyp2e1* ^15^: increased (*p* < 0.05) Expression of *G6pc* and *Ppck1*: higher (*p* < 0.05) Expression of *Srebp1*, *Srepb2*, *Abca1*, *Cpt1*, and *Acc*: higher (*p* < 0.05) triglyceride, LDL ^16^ and FFA ^17^: increased (*p* < 0.05) Expression of *Il1* and *Il6*: n.s. ^2^ N° of microglia and astrocytes: significantly increased (*p* < 0.05)	Exposure altered glucose homeostasis, influenced peripheral Lipid metabolism, and induced hypothalamic inflammation and neuroinflammatory, in a sex-related way.
Badham et al., 2010 [52]	Benzene Prenatal exposure	Intraperitoneal injections: 200 mg/kg, or 400 mg/kg on GD ^18^ 8, 10, 12 and 14	Pregnant CD-1 mice	Offspring: t, t-muconic acid, hydroquinone and catechol	hydroquinone: (fetuses): increased (*p* < 0.01) Catechol: (fetuses): decreased (*p* < 0.01)	Fetuses sex differences in benzene metabolization
Koshko et al., 2021 [53]	Benzene Prenatal exposure	Exposure by inhalation: 50 ppm for 6 h/day from GD 0.5 to GD 21	pregnant C57BL/6JB dams	Offspring: Blood glucose Insulin secretion Insulin resistance	Blood glucose: increased (*p* < 0.05) Insulin secretion: decreased (*p* < 0.05) Insulin resistence: increased (*p* < 0.05)	Insulin resistance in male but not in female offspring

Note: Acute exposure: chemical exposure for 14 days or less; Sub-acute exposure: following a treatment period of 2–4 weeks; Sub-chronic exposure: following a treatment period of 4 weeks–3 months; Chronic exposure: treatment period of 2 years in rat and 18 months in mice [54]. ^1^ SCE: sister chromatid exchanges; ^2^ n.s.: not significant; ^3^ BM: bone marrow; ^4^ CA: chromosomal aberrations; ^5^ MN: micronuclei; ^6^ PHE: Phenol; ^7^ MN-PCE: micronucleated polychromatic erythrocytes; ^8^ MN-NCE: micronucleated normochromatic erythrocytes; ^9^ PCE: polychromatic erythrocytes; ^10^ PCV: Packed Cell Volume; ^11^ Vmax: optimized rate of metabolism; ^12^ t_1/2_: half-life; ^13^ CFU-e: numbers of erythroid colony-forming units; ^14^ HSCs: hematopoietic stem cells; ^15^ Cyp2e1: gene expression of Cytochrome P450 2E1; ^16^ LDL: low-density lipoprotein; ^17^ FFA: non-esterified fatty acid. ^18^ GD: gestational day.

**Table 2 ijerph-19-02339-t002:** Evidence linked to sex differences following in vivo benzene exposure.

		Male	Female
Genotoxicity and Hematotoxicity	BM ^1^ cellular proliferation	+	=Controls
	MN ^2^ in BM cells	++	+
	SCE ^3^	+	+
	MN-PCE ^4^	++	+
	MN-NCE ^5^	++	+
	% PCE ^6^	−−	-
	PCV ^7^	−−	-
	CFU-e numbers ^8^	−	=Controls
Metabolism	HQ ^9^ glucuronide concentration	++	+
	Muconic acid concentration	++	+
	Blood PHE ^10^ concentration	+	++
	Vmax ^11^	++	+
Prenatal exposure	HQ	++ (fetuses)	+ (fetuses)
	Cathecol	++ (fetuses)	+ (fetuses)
	oxidative stress	++ (fetuses)	+ (fetuses)
	MN	++ (fetuses)	+ (fetuses)
	Insulin resistance	++ (offspring)	=Controls

+/−: Higher/Low with respect to controls; ++/−−: Higher/Low with respect to the other sex and controls; =Controls: Not different with respect to controls. ^1^ BM: bone marrow; ^2^ MN: micronuclei; ^3^ SCE: sister chromatid exchange; ^4^ PCE: polychromatic erythrocytes; ^5^ NCE: normochromatic erythrocytes; ^6^ PCE: polychromatic erythrocytes; ^7^ PCV: Packed Cell Volume; ^8^ CFU-e number: numbers of erythroid colony-forming units; ^9^ HQ: Hydroquinone; ^10^ PHE: Phenol; ^11^ Vmax: optimized rate of metabolism.

**Table 3 ijerph-19-02339-t003:** Outcomes of studies on humans related to benzene exposure and sex difference.

Reference	Type of Survey	Interest of the Study	N of Subjects	Outcome
Poynter et al. 2017 [60]	Case/control	Risk of acute myeloid leukemia (AML) and myelodysplastic syndromes (MDS)	265 cases of MDS (180 males/85 females), 420 cases of AML (249 males/171 females) and 1388 controls (773 males/615 females) Benzene exposed: 38 males/5 females with MDS; 58 males/16 females with AML	The risk index about long term benzene exposure (more than 5 years) and AML onset was comparable between males (OR = 1.60, 95% CI 1.04–2.45) and females (OR = 2.84, 95% CI 0.96–8.39), but with higher values in women. For MDS in males OR = 4.30 (95% CI 1.62–11.39) was obtained while the low number of female cases did not permit an OR calculation
Oiamo et al. 2013 [61]	Cross sectional	Sex differences in cardinal symptoms of exposure to chemicals	804 subjects (364 males/440 females)	Women reported five or more symptoms of exposure to BTEX1, NO_2_ and SO_2_, with greater probability (more than 50%) than men
Wang et al., 2021 [62]	Cross sectional	Analysis of benzene exposure on hematotoxicity in workers	218.061 workers	Women showed out of range values of blood parameters (white blood cell, neutrophil count and platelets) different to men, in exposed workers to benzene. Increased S-phenylmercapturic acid (+555%) together with higher DNA damage, including 8-hydroxy-2′-deoxyguanosine (+183%) and malondialdehyde (+33.3%) found in benzene-exposed female workers. Overall, women had a greater risk of hematotoxicity due to benzene exposure.
Moro et al. 2017 [63]	Exposure study (exposed/not exposed)	To assess the possible role of gender on occupational biomarkers of benzene	40 exposed workers (20 men/20 women) and 40 not exposed subjects	Both exposed workers’ groups, male and female, showed higher values of benzene exposure than non-exposed. Even with similar airborne benzene levels the female workers’ group presented increased median t,t,muconic acid/creatinine concentrations compared to male group. Nevertheless, no statistical differences (*p* > 0.05) between sexes were found in genotoxic biomarkers (micronuclei assay, comet test). Erythrocytes, hemoglobin and hematocrit percentage were higher in male-exposed workers than females (*p* < 0.05), while platelets were decreased in exposed men than women
Li et al. 2019 [64]	Cross-sectional	To evaluate how an exposure to volatile organic compounds, at low concentrations but for long time, could produce health effects	499 non occupational exposed subjects	No significant difference was found in blood benzene concentrations between sexes (*p* > 0.05). The higher values of benzene in women were linked to increased urinary white blood cells by 2.902% (95% CI 1.275–6.601)
Li et al. 1994 [65]	Exposure study (exposed/not exposed)	Risk of leukemia and hematopoietic and lymphoproliferative disorders	74.828 benzene exposed workers vs. 35.805 unexposed workers	Evidence about sex differences in mortality, both for cancer and for all type of causes, in the cohort of exposed workers, did not emerge.
Brown et al., 1998 [41]	Physiologically based pharmacokinetics modelling	Identifying differences in internal intake of benzene by sexes	5 male and 5 female subjects	The findings demonstrated that, even considering similar exposure situation, benzene blood concentrations were usually greater in men, but women metabolized more benzene than men (+23–26%) at the same time.
Santiago et al., 2014 [66]	Cross-sectional	Frequencies of chromosomal abnormalities in workers exposed to BTEX regarding sex	50 men and 10 women	Results highlighted a higher risk of chromosomal abnormalities for women exposed to BTEX, even if with a weak statistical significance (*p* = 0.052).

BTEX: benzene, toluene, ethylene, xylenes.

**Table 4 ijerph-19-02339-t004:** Evidence linked to sex differences and benzene exposure in humans.

		Male	Female
Risk of Acute Myeloid Leukemia		OR ^1^ = 1.60 ++	OR ^1^ = 2.84 ++
Cardinal symptoms of exposure			++
Blood parameters	WBC ^2^		++
	Neutrophil count		++
	Platelets	−−	++
	RBC ^3^	++	
	Hb ^4^	++	
	Hc ^5^	++	
Oxidative DNA damage	8-oxod-Gua ^6^		++
Urine parameters	S-phenylmercapturic acid		++
	t,t-muconic acid		++
	leukocyte count		++
Benzene metabolism	Blood benzene	++	
	metabolized benzene		++

^1^ OR Odd Ratio; ^2^ WBC white blood cell, ^3^ RBC red blood cell, ^4^ Hb Hemoglobin, ^5^ Hc Hematocrit, ^6^ 8-oxodGua, 8-hydroxy-2′-deoxyguanosine. ++/−−: Higher/Low with respect to the other sex and controls.

## Data Availability

No new data were created or analyzed in this study. Data sharing is not applicable to this article.

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
