# Peer review of "Sex Difference and Benzene Exposure: Does It Matter?"

_ijerph, 2022, doi:10.3390/ijerph19042339_

Round 1

Reviewer 1 Report

The authors use “Moher, D.; Liberati, A.; Tetzlaff, J.; Altman, D.G. Linee guida per il reporting di revisioni sistematiche e meta analisi: il PRISMA 220 statement. Evidence 2015, 7(6): e1000114.”. In 2020 PISMA was updated regarding reporting guidance for systematic reviews that reflects advances in methods to identify, select, appraise, and synthesise studies.

The selection of the articles was performed using any app? For example through Rayyan?

Author Response

Response to referee 1

Comments and Suggestions for Authors

The authors use “Moher, D.; Liberati, A.; Tetzlaff, J.; Altman, D.G. Linee guida per il reporting di revisioni sistematiche e meta analisi: il PRISMA 220 statement. Evidence 2015, 7(6): e1000114.”. In 2020 PISMA was updated regarding reporting guidance for systematic reviews that reflects advances in methods to identify, select, appraise, and synthesise studies.

The selection of the articles was performed using any app? For example through Rayyan?

  1. We thank the referee for the suggestion, as written in the paper, we use web scientific database (Scopus, pubmed) but no app.

We certainly take into consideration this suggestion for future review.

Reviewer 2 Report

The work brings essential information for the field with such important metanalysis showing the sex-manner effects of benzene in rodents and humans. However, there is a literature gap about those topics, and it would be a significant contribution to having a review paper discussing this topic. However, many structural issues specified in the sequence must be fixed. 

1- The following sentences were entirely copied from the original paper. “The analysis of the biology of sex differences as well as the development of gender-specific medicine is a milestone in the advancement of our knowledge in different fields of life sciences.” Ortona E, Delunardo F, Baggio G, Malorni W. A sex and gender perspective in medicine: a new mandatory challenge for human health. Preface. Ann Ist Super Sanita. 2016 Apr-Jun;52(2):146-8. doi: 10.4415/ANN_16_02_02. PMID: 27364385.

2- The authors talk of two different conceptual words in the first paragraph, sex, and gender. However, the whole paper expresses an analysis based on the biological characteristics of sex. Discussing the gender role should be such a more complex analysis. It should include the psychological, cultural, and social context of those humans involved in the study and how that interferes with benzene exposure. For example, the survey should include where men and women usually work and how that interferes with the way of being exposed. Is it by water? Detergent? Paint? Gas station? Smoking? Industry?. Considering that most studies are based on biological analysis, the authors must consider changing the title to sex instead of gender.

3- The following paragraph should be on the introduction “The Committee on Understanding the Biology of Sex and Gender Differences of the US Institute of Medicine distinguished sex and gender with the following definitions: “sex is the classification of living things generally as either male or female, according to their reproductive organs and functions assigned by the chromosomal complement” while gender is “a person’s  self-representation as male or female, or how that person is responded to by social institutions…” [58].”

4- There are many English issues, for example, in figure 1. (11 records identified “throught” other ways). Therefore, it is recommended to submit to a specialized company.

5- The paragraphs on results are disconnected. Write a sentence that begins by introducing the new topic for that section and connects readers back to the main idea of the piece of writing. At the end of each paragraph, It would be essential to emphasize the mechanisms suggested by the authors about the sex differences observed.

6- In addition, as mentioned before, there are many phrases not in the appropriate language, and it is not very technical. It will be essential to suggest the differences in their approaches that can justify differences. Please, modify the following sentences:  -“These results were then confirmed by the same authors.” - “However, not all studies agree. - “ Finally” - “These data were confirmed in a later paper where they showed.” - The same year, they studied [41]” - “Finally.” 

7- On the subtitle, 3.2. Gender difference and benzene exposure: findings in humans. It must be considered writing those findings that were added to the table. Connecting those data and discussing their differences would be essential. 

8- Table 3a and Table 3b must be on results.

9- The metabolism of benzene and figure 2 mentioned in the discussion must be in the introduction 

10- The discussion section focuses too much on genotoxicity and hematotoxicity induced by benzene. Therefore, the paragraphs should be shorter and avoid describing results again. Instead, explain the main idea of each study and interconnect with other authors, bring methodology limitations and possible mechanisms, and offer new hypotheses. The physiological differences between males and females should also be included in this context, for example, the role of gonadal hormones on inflammation.  

11- The conclusion must be rewritten. Some parts should be in the introduction, and some elements should be in the discussion section. It must have a maximum of two paragraphs and be objective. 

12- The following papers would contribute to the paper analysis. It shows data in pregnant women, children, adults, and the elderly. Having those data in different developmental phases should give insights into the discussion related to physiology and gonadal hormones.

  • Amin MM, Rafiei N, Poursafa P, Ebrahimpour K, Mozafarian N, Shoshtari-Yeganeh B, Hashemi M, Kelishadi R. Association of benzene exposure with insulin resistance, SOD, and MDA as markers of oxidative stress in children and adolescents. Environ Sci Pollut Res Int. 2018 Dec;25(34):34046-34052. doi: 10.1007/s11356-018-3354-7. Epub 2018 Oct 3. PMID: 30280344.
  • Park HS, Seo JC, Kim JH, Bae SH, Lim YH, Cho SH, Hong YC. Relationship Between Urinary t, t-muconic Acid and Insulin Resistance in the Elderly. Korean J Occup Environ Med. 2011;23(4):387-396.
  • Shim, Y.H.; Ock, J.W.; Kim, Y.-J.; Kim, Y.; Kim, S.Y.; Kang, D. Association between Heavy Metals, Bisphenol A, Volatile Organic Compounds and Phthalates and Metabolic Syndrome. J. Environ. Res. Public Health201916, 671. https://doi.org/10.3390/ijerph16040671.
  • Andrew D Williams, Katherine L Grantz, Cuilin Zhang, Carrie Nobles, Seth Sherman, Pauline Mendola, Ambient Volatile Organic Compounds and Racial/Ethnic Disparities in Gestational Diabetes Mellitus: Are Asian/Pacific Islander Women at Greater Risk?, American Journal of Epidemiology, Volume 188, Issue 2, February 2019, Pages 389–397, https://doi.org/10.1093/aje/kwy256
  • Kelishadi R, Mirghaffari N, Poursafa P, Gidding SS. Lifestyle and environmental factors associated with inflammation, oxidative stress and insulin resistance in children. Atherosclerosis. 2009 Mar;203(1):311-9. doi: 10.1016/j.atherosclerosis.2008.06.022. Epub 2008 Jul 1. PMID: 18692848.
  • Choi SH, Kwon TG, Chung SK, Kim TH. Surgical smoke may be a biohazard to surgeons performing laparoscopic surgery. Surg Endosc. 2014 Aug;28(8):2374-80. doi: 10.1007/s00464-014-3472-3. Epub 2014 Feb 26. PMID: 24570016.

Minor:

 1- The long form of the MN-NCE abbreviation is missing in section 3.1.1. Genotoxicity and hematoxicity. In addition, there is a typo in this subtitle, and the correct way is hematotoxicity.

Author Response

Response to referee 2

Comments and Suggestions for Authors

The work brings essential information for the field with such important metanalysis showing the sex-manner effects of benzene in rodents and humans. However, there is a literature gap about those topics, and it would be a significant contribution to having a review paper discussing this topic. However, many structural issues specified in the sequence must be fixed. 

  • The following sentences were entirely copied from the original paper. “The analysis of the biology of sex differences as well as the development of gender-specific medicine is a milestone in the advancement of our knowledge in different fields of life sciences.” Ortona E, Delunardo F, Baggio G, Malorni W. A sex and gender perspective in medicine: a new mandatory challenge for human health. Preface. Ann Ist Super Sanita. 2016 Apr-Jun;52(2):146-8. doi: 10.4415/ANN_16_02_02. PMID: 27364385.
  1. We apologize, the text is now changed
  • The authors talk of two different conceptual words in the first paragraph, sex, and gender. However, the whole paper expresses an analysis based on the biological characteristics of sex. Discussing the gender role should be such a more complex analysis. It should include the psychological, cultural, and social context of those humans involved in the study and how that interferes with benzene exposure. For example, the survey should include where men and women usually work and how that interferes with the way of being exposed. Is it by water? Detergent? Paint? Gas station? Smoking? Industry?. Considering that most studies are based on biological analysis, the authors must consider changing the title to sex instead of gender.
  1. We really thank the referee for this indication; the text is now changed taken into consideration this suggestion.
  • The following paragraph should be on the introduction “The Committee on Understanding the Biology of Sex and Gender Differences of the US Institute of Medicine distinguished sex and gender with the following definitions: “sexis the classification of living things generally as either male or female, according to their reproductive organs and functions assigned by the chromosomal complement” while gender is “a person’s  self-representation as male or female, or how that person is responded to by social institutions…” [58].”

  1. According to the reviewer’s suggestion, this paragraph is now in the Introduction.

  • There are many English issues, for example, in figure 1. (11 records identified “throught” other ways). Therefore, it is recommended to submit to a specialized company.

  1. We are sorry for the typo, the figure 1 is now correctly changed and the text has been reviewed for English.

  • The paragraphs on results are disconnected. Write a sentence that begins by introducing the new topic for that section and connects readers back to the main idea of the piece of writing. At the end of each paragraph, It would be essential to emphasize the mechanisms suggested by the authors about the sex differences observed.
  1. According to the reviewer’s suggestion, this section has been modified.
  • In addition, as mentioned before, there are many phrases not in the appropriate language, and it is not very technical. It will be essential to suggest the differences in their approaches that can justify differences. Please, modify the following sentences:  -“These results were then confirmed by the same authors.” - “However, not all studies agree. - “ Finally” - “These data were confirmed in a later paper where they showed.” - The same year, they studied [41]” - “Finally.” 

  1. These “opening words”, often read in scientific papers, have been used as a stylistic choice to make the reading easier, avoiding a mere list of the results. However, where possible, we have tried to make the speech drier, even if we do not believe that these “incipits” affect the technical/scientific quality of the work.

Changes made are highlighted in the text.

  • On the subtitle, 2. Gender difference and benzene exposure: findings in humans. It must be considered writing those findings that were added to the table. Connecting those data and discussing their differences would be essential. 
  1. According to the reviewer’s suggestion, this section has been modified.
  • Table 3a and Table 3b must be on results.
  1. These tables have been moved on the result section. Table 3a is now Table 2 and table 3b is now Table 4.
  • The metabolism of benzene and figure 2 mentioned in the discussion must be in the introduction 
  1. These changes have been made: Figure 2 is now Figure 1.
  • The discussion section focuses too much on genotoxicity and hematotoxicity induced by benzene. Therefore, the paragraphs should be shorter and avoid describing results again. Instead, explain the main idea of each study and interconnect with other authors, bring methodology limitations and possible mechanisms, and offer new hypotheses.

  1. According to the reviewer’s suggestion, the discussion has been shortened and it has been also focused on non-cancer benzene effects.

The physiological differences between males and females should also be included in this context, for example, the role of gonadal hormones on inflammation.  

  1. The physiological differences between sexes are described in the introduction section. However, some consideration about the possible role of the endocrine system have been added in discussion.

 The following papers would contribute to the paper analysis. It shows data in pregnant women, children, adults, and the elderly. Having those data in different developmental phases should give insights into the discussion related to physiology and gonadal hormones.

  1. The aim of the study is clarifying the possible role of sex in the health effects due to benzene exposure, starting from animal models to epidemiological surveys. We have chosen to select only scientific papers that considered both sexes, while studies that evaluated only one sex have been excluded. This choice, if on the one hand reduces the number of the scientific paper selected, on the other hand limits/reduces the bias of the results, because males/females are in the same conditions.

Secondly, since genotoxic and haematotoxic effects of benzene have been the most studied, the results related to sex differences are almost exclusively related to these issues. There has been an increased interest over the years for non-cancer effects of benzene (e.g. insulin resistance, diabetes mellitus, metabolic syndrome), as suggested by the reviewer. However, in most cases, these papers (including those suggested by the reviewer) are not focused on the difference between male and female or describe not conclusive/preliminary results. Moreover, with regard the mechanisms involved in non-carcinogenic effects of benzene, the hypotheses proposed are few and rarely consider the difference between the sexes.

However, the text is now changed with further considerations and other bibliographic references about other health effects following benzene exposure, also considering different stage of life.

With regard to the following articles suggested by the reviewer,

  • Park HS, Seo JC, Kim JH, Bae SH, Lim YH, Cho SH, Hong YC. Relationship Between Urinary t, t-muconic Acid and Insulin Resistance in the Elderly. Korean J Occup Environ Med. 2011;23(4):387-396.

Neither abstract nor full paper were found. We asked the authors for the article without receiving reply.  Therefore, we decided to consider their subsequent publication reporting the association between benzene exposure and insulin resistance in elderly adults in participants of the Korean Elderly Environmental Panel (KEEP) study:

Choi Y-H, Kim JH, Lee B-E, Hong Y-C (2014) Urinary benzene metabolite and insulin resistance in elderly adults. Sci Total Environ 482: 260–268.

  • Kelishadi R, Mirghaffari N, Poursafa P, Gidding SS. Lifestyle and environmental factors associated with inflammation, oxidative stress and insulin resistance in children. Atherosclerosis. 2009 Mar;203(1):311-9. doi: 10.1016/j.atherosclerosis.2008.06.022. Epub 2008 Jul 1. PMID: 18692848.

In this work, environmental air pollution was referred to PM10, ozone, sulfur dioxide, nitrogen dioxide, and carbon monoxide, while benzene was not mentioned. Urinary benzene metabolites (e.g. t,t-muconic acid) were not measured in children cohort enrolled. Therefore, this article is not focused on benzene exposure. However, we have considered the general aim of the paper because benzene is a known environmental pollutant, associated with the chemical agents listed above.

  • Choi SH, Kwon TG, Chung SK, Kim TH. Surgical smoke may be a biohazard to surgeons performing laparoscopic surgery. Surg Endosc. 2014 Aug;28(8):2374-80. doi: 10.1007/s00464-014-3472-3. Epub 2014 Feb 26. PMID: 24570016.

The reviewer recommended adding further data on benzene exposure and non-cancer effect due to environmental pollution considering changes (e.g hormone levels) and the relative mechanisms that can happen at different stages in life: pregnant women, children, adults, and the elderly. However, this work evaluates the surgical smoke as a cancer risk to surgeons performing laparoscopic surgery. In fact, surgical smoke contains several chemical agents, including benzene, for which the cancer risk was classified as unacceptable. Therefore, we believe this article is not relevant based on the auditor’s comment.

  • Amin MM, Rafiei N, Poursafa P, Ebrahimpour K, Mozafarian N, Shoshtari-Yeganeh B, Hashemi M, Kelishadi R. Association of benzene exposure with insulin resistance, SOD, and MDA as markers of oxidative stress in children and adolescents. Environ Sci Pollut Res Int. 2018 Dec;25(34):34046-34052. doi: 10.1007/s11356-018-3354-7. Epub 2018 Oct 3. PMID: 30280344.
  • Shim, Y.H.; Ock, J.W.; Kim, Y.-J.; Kim, Y.; Kim, S.Y.; Kang, D. Association between Heavy Metals, Bisphenol A, Volatile Organic Compounds and Phthalates and Metabolic Syndrome.  Environ. Res. Public Health201916, 671. https://doi.org/10.3390/ijerph16040671.
  • Andrew D Williams, Katherine L Grantz, Cuilin Zhang, Carrie Nobles, Seth Sherman, Pauline Mendola, Ambient Volatile Organic Compounds and Racial/Ethnic Disparities in Gestational Diabetes Mellitus: Are Asian/Pacific Islander Women at Greater Risk?, American Journal of Epidemiology, Volume 188, Issue 2, February 2019, Pages 389–397, https://doi.org/10.1093/aje/kwy256

These references are now added in the text, together with others.

The conclusion must be rewritten. Some parts should be in the introduction, and some elements should be in the discussion section. It must have a maximum of two paragraphs and be objective. 

  1. The conclusion has been modified following the reviewer’s suggestion.

Minor:

 1- The long form of the MN-NCE abbreviation is missing in section 3.1.1. Genotoxicity and hematoxicity. In addition, there is a typo in this subtitle, and the correct way is hematotoxicity

  1. We apologize for the typo. The required changes has been made.

Reviewer 3 Report

Introduction

It would be useful for the authors to define or state the differences between the two words "sex" and "Gender" . these words are used interchangeably in the article , nonetheless in epidemiological studies these terms does not mean the same thing and may play different roles in exposure science.

Recent researches (9) shows that ......."The authors should cite more than one study in this case to support this statement instead of only providing one reference.

Some sentences in the beginning of the introduction are missing references.  e.g. Women had a prevalence of repetitive etc...please provide a reference.

Rate is reported as an odds ration for the study referred to (refence 10). Please check what measure of effect was used in this study and modify/revise sentence (rate ratio or Odds ratio). 

Under the paragraph Clewell et al...Were the results on absorption phase statistically proven? if yes please provide P values and a reference at the end of the paragraph.

The entire paragraph is not refence. Please provide appropriate references.

Please provide references for the first paragraphs on Benzene.

Provide references for the statement; A very large number of studies concerning DNA  etc.

The aim is not clear and should be revised. the study is a systematic review of published studies . this should be included in the aim.

Methods

Was there any reason sex was not included as a search term? in the review sex and gender are both used.

Please describe the study quality indicators used in the systematic review. that is, how quality of the studies was assessed and by which methods. Provide a reference and a brief description of the method.

2.3 Bias assessment

Please describe using a table how this assessment was done- the authors indicated using a funnel plot. At the moment it is difficult to understand this section.

Results

The references / studies presented on the table 1 are outdated. Please update and use most recent references. 

3.1.1 Genotoxicity 

Please provide references or source of information  for the sentences on DNA damage response in the paragraph.

Most of the references or studies in the paper (Table 1) do not describe the methods used for their analysis- this is a limitation especially when comparing two results. The authors should go back to the studies and include the methods of analysis used by each study.

Table 2: In the first reference, the measure of effect Odds ration is not interpreted as it should. Please revise.

Were the authors indicate a significant or no significant difference a Pvalue or 95%CI of the data should be provided. This should apply when using statements such as higher risks  (please provide a risk ration or hazard ratio) of the results. 

Conclusions

When doing a systematic review with toxicology studies and epidemiology studies on substance exposure its better to find the combined effect based on the sourced paper. This way on the conclusion the authors are able to indicate without a doubt the  effect of the investigated substances.

The authors did not provide adequate references for the studies used in the paper, in the introduction as well as the discussion. More recent publications or literature may be sources to get a feel on where the topic is at present. 

Extensive English editing and spelling corrections are required. 

The paper is more of a scoping review rather than a systematic review. I would advice to rethink the title.

Author Response

Response to Referee n.3

Comments and Suggestions for Authors

Introduction

It would be useful for the authors to define or state the differences between the two words "sex" and "Gender". These words are used interchangeably in the article, nonetheless in epidemiological studies these terms does not mean the same thing and may play different roles in exposure science.

  1. We really thank the referee for this indication; the text has been changed taken into consideration this suggestion.

Recent researches (9) shows that ......."The authors should cite more than one study in this case to support this statement instead of only providing one reference.

  1. The reference has been added.

Some sentences in the beginning of the introduction are missing references.  e.g. Women had a prevalence of repetitive etc...please provide a reference.

  1. The reference has been added

Rate is reported as an odds ration for the study referred to (refence 10). Please check what measure of effect was used in this study and modify/revise sentence (rate ratio or Odds ratio). 

  1. The cited paper studied if female workers have a higher risk of injury compared with males when performing the same job, and evaluated sex differences in the type or severity of injury. A multivariate logistic regression (adjusted for job, tenure and age) was used to calculate odds ratios and 95% confidence intervals for female versus male injury risk for all injuries, recordable injuries and lost work time injuries. Then the analysis was repeated for acute injuries and musculoskeletal disorder. This aspet has been better clarified in the text.

Under the paragraph Clewell et al...Were the results on absorption phase statistically proven? if yes please provide P values and a reference at the end of the paragraph.

  1. As written, Clewell et al produced a review of published articles on the theme of “…Potential Impact of Age- and Gender-Specific Pharmacokinetic Differences on Tissue Dosimetry”, where there are no statistically references (p value) but only other bibliographic references, in case.

The entire paragraph is not reference. Please provide appropriate references.

  1. These references have been added, in addition to the 27 already cited.

Please provide references for the first paragraphs on Benzene.

  1. References have been added

Provide references for the statement; A very large number of studies concerning DNA  etc.

  1. The sentence has been deleted

The aim is not clear and should be revised. the study is a systematic review of published studies . this should be included in the aim.

  1. According to reviewer’s suggestion, the aim has been better clarified.

Methods

Was there any reason sex was not included as a search term? in the review sex and gender are both used.

  1. We really thank the referee for this indication: the term sex was included and the literature search implemented and the text was changed taken into consideration the difference between the meaning of “sex” and “gender”.

Please describe the study quality indicators used in the systematic review. that is, how quality of the studies was assessed and by which methods. Provide a reference and a brief description of the method.

  1. We decided to follow the referee's suggestion and to consider the work as a scoping review; the searching strategy, inclusion criteria and all information have been described in the text. Since it is now a scoping review, the study quality indicators of the systematic review are now not applicable.

2.3 Bias assessment

Please describe using a table how this assessment was done- the authors indicated using a funnel plot. At the moment it is difficult to understand this section.

R.In method, we wrote about Fannel Plot, that was impossible to proceed with its realization:

“Even if the statistical assessment of publication bias was planned, using a Funnel plot, it was not possible to carried out it due to the conditions of using Funnel plots not being met, especially insufficient number of studies and heterogeneity between studies.”

While for “The assessment of bias within studies was carried out using the Cochrane risk of bias tool [29]”

However, now, considering a scoping review, as suggested by the referee, it is not necessary to explained the bias assessment, so the text has been changed and simplified.

Results

The references / studies presented on the table 1 are outdated. Please update and use most recent references. 

  1. We have chosen to select only scientific papers that considered both sexes, while studies that evaluated only one sex have been excluded. This choice, if on the one hand reduces the number of the scientific paper selected, on the other limits/reduces the bias of the results, because males/females are in the same experimental conditions. In addition, as outlined in discussion, most of this research is from the 1980s and 1990s, although in recent years the interest in this field was increasing.

However, to have a more complete scenario, we decided to include also benzene prenatal exposure, which de facto showed sex-based differences. The selected articles are more recent.

3.1.1 Genotoxicity 

Please provide references or source of information  for the sentences on DNA damage response in the paragraph.

  1. The reference was Faiola et al, [21], however the sentence is now changed and the reference is more clear.

Most of the references or studies in the paper (Table 1) do not describe the methods used for their analysis- this is a limitation especially when comparing two results. The authors should go back to the studies and include the methods of analysis used by each study.

  1. Sorry, but we disagree. The selected papers are very different, from the study design to the outcomes. The interest is in the consideration of each study results, because a comparison among studies is somewhat not useful. Only for few papers (i.e. the analysis of MN) we have the same target (but with different samples, different benzene exposure situations, different sampling time, and so on…) but the interest of the presented review is not for the single analytical data but for the consideration about differences between sexes, for this reason the only way to avoid bias of different methodological approach is considering the findings of single study and not a comparison.

From this point of view the addition of information about “analytical” methods seems a burden of the table and of the data provided without giving an added value to the discussion. Nevertheless, a general consideration about the different methodological approach is now added in the text.

Table 2: In the first reference, the measure of effect Odds ration is not interpreted as it should. Please revise.

  1. The sentence has been changed.

Were the authors indicate a significant or no significant difference a Pvalue or 95%CI of the data should be provided. This should apply when using statements such as higher risks  (please provide a risk ration or hazard ratio) of the results. 

  1. The text has been changed and, where indicated, information has been added.

Conclusions

When doing a systematic review with toxicology studies and epidemiology studies on substance exposure its better to find the combined effect based on the sourced paper. This way on the conclusion, the authors are able to indicate without a doubt the effect of the investigated substances.

  1. Currently it is a scoping review, so some considerations have been changed.

We selected studies maily focused on exposure to benzene. The articles that have considered the exposure to more substances, mostly are related to environmental pollution/exposure. However, the results related to benzene exposure have been associated to benzene considering specific exposure data (e.g.,biological indicators of benzene exposure). All these aspects have been better clarified.

In discussion, a consideration has been added regarding the limits related to benzene environmental exposure, due to its association with other xenobiotics.

The authors did not provide adequate references for the studies used in the paper, in the introduction as well as the discussion. More recent publications or literature may be sources to get a feel on where the topic is at present. 

  1. The literature search has been implemented, and references were added.

Extensive English editing and spelling corrections are required. 

  1. The English has been revised.

The paper is more of a scoping review rather than a systematic review. I would advice to rethink the title.

  1. Thank you for this suggestion, we agree. The text is now changed according to this item.

Round 2

Reviewer 2 Report

The work shows such interesting metanalysis about the effects of benzene in a sex-manner in rodents and humans. Moreover, the researchers have been substantially addressed the comments, and they made significant improvements in the writing, grammar, and structural text organization.

Minor comments:

  1. There are seven misspellings of the word insulin. It was written as insuline.
  2. On page 12, please change the part “metabolic lost balance” to metabolic imbalance.
  3. On-Page 4 of the discussion, line 160, there is a citation {57) written to be removed.
  4. On discussion, lines 173-174, please remove the following phrase: Researchers [79] underlined that, about pharmacokinetic findings, the most correct extrapolation to predict human clearance is by monkey while the use of dog is the worst.

Author Response

Minor comments:

There are seven misspellings of the word insulin. It was written as insuline.

The words have been corrected.

On page 12, please change the part “metabolic lost balance” to metabolic imbalance.

The correction required has been made.

On-Page 4 of the discussion, line 160, there is a citation {57) written to be removed.

This typo has been removed.

On discussion, lines 173-174, please remove the following phrase: Researchers [79] underlined that, about pharmacokinetic findings, the most correct extrapolation to predict human clearance is by monkey while the use of dog is the worst.

This sentence has been deleted.

Reviewer 3 Report

The authors seemed to addressed all the comments from the review report except one comments they disagreed with (below):

Most of the references or studies in the paper (Table 1) do not describe the methods used for their analysis- this is a limitation especially when comparing two results. The authors should go back to the studies and include the methods of analysis used by each study. Sorry, but we disagree. The selected papers are very different, from the study design to the outcomes. The interest is in the consideration of each study results, because a comparison among studies is somewhat not useful. Only for few papers (i.e. the analysis of MN) we have the same target (but with different samples, different benzene exposure situations, different sampling time, and so on…) but the interest of the presented review is not for the single analytical data but for the consideration about differences between sexes, for this reason the only way to avoid bias of different methodological approach is considering the findings of single study and not a comparison. From this point of view, the addition of information about “analytical” methods seem a burden of the table and of the data provided without giving an added value to the discussion. Nevertheless, a general consideration about the different methodological approach is now added in the text.

Overall, I am happy with the revised article. Perhaps the authors can add this to the limitation section- that comparison of the reviews were based on findings and different analytical methods were not considered.

Author Response

The authors seemed to addressed all the comments from the review report except one comments they disagreed with (below):

Most of the references or studies in the paper (Table 1) do not describe the methods used for their analysis- this is a limitation especially when comparing two results. The authors should go back to the studies and include the methods of analysis used by each study. Sorry, but we disagree. The selected papers are very different, from the study design to the outcomes. The interest is in the consideration of each study results, because a comparison among studies is somewhat not useful. Only for few papers (i.e. the analysis of MN) we have the same target (but with different samples, different benzene exposure situations, different sampling time, and so on…) but the interest of the presented review is not for the single analytical data but for the consideration about differences between sexes, for this reason the only way to avoid bias of different methodological approach is considering the findings of single study and not a comparison. From this point of view, the addition of information about “analytical” methods seem a burden of the table and of the data provided without giving an added value to the discussion. Nevertheless, a general consideration about the different methodological approach is now added in the text.

Overall, I am happy with the revised article. Perhaps the authors can add this to the limitation section- that comparison of the reviews were based on findings and different analytical methods were not considered.

This item has been added both in “2.2. Eligibility criteria and study selection” and in Discussion.
